# Effects of Coenzyme Q10 on the Biomarkers (Hydrogen, Methane, SCFA and TMA) and Composition of the Gut Microbiome in Rats

**DOI:** 10.3390/ph16050686

**Published:** 2023-05-02

**Authors:** Anastasiia Yu. Ivanova, Ivan V. Shirokov, Stepan V. Toshchakov, Aleksandra D. Kozlova, Olga N. Obolenskaya, Sofia S. Mariasina, Vasily A. Ivlev, Ilya B. Gartseev, Oleg S. Medvedev

**Affiliations:** 1Faculty of Medicine, Lomonosov Moscow State University, Moscow 119991, Russia; 2National Medical Research Center of Cardiology of the Ministry of Health of the Russian Federation, Laboratory of Experimental Pharmacology, Moscow 121552, Russia; 3Medical and Technical Information Technologies, Bauman Moscow State Technical University, Moscow 105005, Russia; 4Center for Genome Research, National Research Center “Kurchatov Institute”, Moscow 123098, Russia; 5Institute of Functional Genomics, Moscow State University, Moscow 119991, Russia; 6Pharmacy Resource Center, Peoples Friendship University of Russia (RUDN University), Moscow 117198, Russia; 7The Institute of Artificial Intelligence of Russian Technological University MIREA, Moscow 119454, Russia

**Keywords:** coenzyme Q10, antioxidant, molecular hydrogen, short-chain fatty acids, trimethylamine, gut microbiota

## Abstract

The predominant route of administration of drugs with coenzyme Q10 (CoQ10) is administration *per os*. The bioavailability of CoQ10 is about 2–3%. Prolonged use of CoQ10 to achieve pharmacological effects contributes to the creation of elevated concentrations of CoQ10 in the intestinal lumen. CoQ10 can have an effect on the gut microbiota and the levels of biomarkers it produces. CoQ10 at a dose of 30 mg/kg/day was administered *per os* to Wistar rats for 21 days. The levels of gut microbiota biomarkers (hydrogen, methane, short-chain fatty acids (SCFA), and trimethylamine (TMA)) and taxonomic composition were measured twice: before the administration of CoQ10 and at the end of the experiment. Hydrogen and methane levels were measured using the fasting lactulose breath test, fecal and blood SCFA and fecal TMA concentrations were determined by NMR, and 16S sequencing was used to analyze the taxonomic composition. Administration of CoQ10 for 21 days resulted in a 1.83-fold (*p* = 0.02) increase in hydrogen concentration in the total air sample (exhaled air + flatus), a 63% (*p* = 0.02) increase in the total concentration of SCFA (acetate, propionate, butyrate) in feces, a 126% increase in butyrate (*p* = 0.04), a 6.56-fold (*p* = 0.03) decrease in TMA levels, a 2.4-fold increase in relative abundance of *Ruminococcus* and *Lachnospiraceae AC 2044 group* by 7.5 times and a 2.8-fold decrease in relative representation of *Helicobacter*. The mechanism of antioxidant effect of orally administered CoQ10 can include modification of the taxonomic composition of the gut microbiota and increased generation of molecular hydrogen, which is antioxidant by itself. The evoked increase in the level of butyric acid can be followed by protection of the gut barrier function.

## 1. Introduction

The Coenzyme Q10 (CoQ10) was first identified by Crane et al. (1957) in the beef heart mitochondria [1]. CoQ10, located in the inner mitochondrial membrane plays a key role in the process of oxidative phosphorylation, and production of ATP, as highlighted by Nobel Prize winner of the 1978, Peter Mitchell. CoQ is composed of a benzoquinone ring and a polyisoprenoid tail containing between 6 and 10 species-specific subunits conferring stability to the molecule inside the phospholipid bilayer. The isoprene chain in *Saccharomyces cerevisiae* contains six subunits (CoQ6), seven subunits are present in *Crucianella maritima* (CoQ7), eight in *E. coli* (CoQ8), nine and ten in mice and rats (CoQ9 and CoQ10), and ten in humans (CoQ10) [2,3]. This molecule has traditionally been used clinically as an enhancer of mitochondrial function or an antioxidant intended to either palliate or diminish the oxidative damage that may worsen the physiological outcome of a wide range of diseases (cardiovascular, endocrine etc.) [3]. The absorption of CoQ10 is slow and limited due to its hydrophobicity and large molecular weight and, therefore, high doses are needed to reach a number of rat tissues [4].

Benzoquinones are synthesized by both aerobic [5] and anaerobic bacteria [6]. The physiological role of ubiquinone in bacteria is the regulation of energy metabolism, gene expression, and prevention of oxidative stress [7]. Benzoquinone compounds have been shown to act as a growth factor for some uncultured bacteria of the gut microbiome [8].

One of the colon microbiota functions is fermentation of complex carbohydrates/fibers that cannot be absorbed in the small intestine. As a result of this process, low-molecular-weight biologically active substances are formed, including gaseous metabolites (hydrogen (H_2_), methane (CH_4_), short-chain fatty acids (SCFA) and trimethylamine (TMA). A group of Japanese scientists showed that molecular hydrogen has an antioxidant effect [9]. H_2_ is able to bind the most aggressive reactive oxygen species—hydroxyl radical (•OH) and peroxynitrite (ONOO^−^), increase the expression of endogenous proteins—superoxide dismutase, catalase, glutathione peroxidase, myeloperoxidase, glutathione synthase; exhibit cytoprotective properties by regulating the Nrf2-, ARE-, PI3K/Akt-, JAK2/STAT3 signaling pathways inside the cell; reduce pro-inflammatory cytokine formation [10,11]. It has been shown that H_2_ administration in the form of hydrogen-saturated water leads to increased formation of CoQ10 in rat heart mitochondria, increased ATP generation and suppression of oxidative stress, which is accompanied by a decrease in malonic dialdehyde (MDA) levels [12]. 

Increased concentrations of CoQ10 in the intestinal lumen when taken orally can create conditions for changes in the composition of the intestinal microbiota, as well as influence the production of biomarkers, which, however, has not been experimentally investigated so far. Therefore, the purpose of this work was to study the effect of oral administration of CoQ10 on both the taxonomic composition of the gut microbiota in rats and the level of such biomarkers of its enzymatic activity as hydrogen, methane, SCFA and TMA.

## 2. Results

### 2.1. Determination of Gas Metabolites of Gut Microbiota (H_2_ and CH_4_)

#### 2.1.1. Determination of H_2_ and CH_4_ in the Rat Total Air Sample on the Regular Diet

The mean values of triplicate measurements of H_2_ and CH_4_ levels (0 h, 3 h, 6 h) in air samples during the day before and at the end of the experiment are presented in Table 1. CH_4_ concentrations in the air sample on the regular diet were significantly higher than H_2_ concentrations in all experimental groups: 7-fold in the *Control* group, 9.7-fold in the *Vehicle* group, 8-fold in the *CoQ10* group. The ratio CH_4_/H_2_ at the end of the experiment increased in the *Control* group by 53.8% and decreased in the *Vehicle* and *CoQ10* groups by 23.03% and 42.06%, respectively. However, no statistically significant differences were found in either group. 

#### 2.1.2. Determination of H_2_ and CH_4_ in the Rat Total Air Sample in the Lactulose Test

The metabolic activity of the gut microbiota was assessed by stimulating the enzymatic activity by administration of the artificial disaccharide lactulose (4-O-β-D-galactopyranosyl-D-fructose) in the absence of food factors. Comparison of the areas under the curves (AUC) of the dynamics of H_2_ and CH_4_ concentration in the total air sample after lactulose loading for a period of 8 h revealed an increase in H_2_ generation in the *CoQ10* group by 1.83 times relative to baseline and was 83.26 ± 9.17 ppm*h (*p* = 0.02). A statistically significant increase in the dynamics of CH_4_ generation was found in the *Control* and *Vehicle* groups 2.16-fold and 1.47-fold, and was 1036 ± 215 ppm*h (*p* = 0.01) and 679.6 ± 91.4 ppm*h (*p* = 0.04), respectively, shown in Figure 1.

### 2.2. Determination of SCFA in Plasma and Feces of Rats

In the *CoQ10* group, acetic acid concentration increased at the end of the experiment by 56% (*p* = 0.04), butyric acid by 126% (*p* = 0.04) in the fecal samples. Total fecal SCFA concentration increased in the *CoQ10* group by 63% compared to baseline values (*p* = 0.02). No statistically significant differences were found in the plasma in all experimental groups. Data is presented in Table 2.

### 2.3. Determination of TMA in the Feces of Rats

The concentration of TMA in the feces of rats in the *CoQ10* group at the end of the experiment was 1.12 ± 0.44 μmol/g and was 6.56 times lower than the baseline level (*p* = 0.03), as shown in Figure 2.

### 2.4. Determination of CoQ10 in Plasma and Feces of Rats

At the end of the experiment, CoQ10 levels in the *CoQ10* group were increased 2.2 times and 5.9 times in plasma and feces, respectively, compared to than its average content in the other experimental groups (*p* < 0.01) (Figure 3).

### 2.5. Analysis of the Gut Microbiota Composition

13,556 amplicon sequence variants (ASVs) out of 917,866 total reads were obtained after sequencing. On average, there were about 6374 reads per replicate. After filtering, trimming and chimera removal with DADA2 R package (REF) there were obtained 1127 ASVs which was associated with 138 genera. ASV rarefaction curves built with the Vegan R package reached the saturation of approximately 1500 reads for all sequenced samples indicating the sufficiency of sequencing depth (Appendix A).

#### 2.5.1. Alpha-Diversity Analysis

The alpha-diversity analysis performed by the microeco package was using the Shannon and Inv. Simpson indexes. Comparison made by t-test showed a significant difference in Shannon index between “pre-post” in the *Vehicle* group (*p* = 0.03) and *CoQ10* group (*p* = 0.03). The Inv. Simpson index increased only in the *CoQ10* group (*p* = 0.0008) (Figure 4A).

Difference between microbial diversity in groups was verifying with conducted a permutational multivariate analysis of variance (ADONIS, “adonis2” function) based on Euclidean dissimilarities and on the data with centroid log ratio by reason of the compositional behaviour of the absolute data (“phyloseq::distance” function). Permanova showed a significant difference in bacteria representatives in different experimental groups (betadisper pV > 0.01−variances are homogenous, adonis2 pV = 0.001). Principal coordinate analysis (PcoA) based on Bray–Curtis dissimilarities on relative normalisation revealed a significant separation of samples according to different experimental groups (Figure 4B).

#### 2.5.2. Microbial Community Composition

The community composition of the gut microbiota in rats was dominated by *Firmicutes* and *Bacteroidota* phyla (Figure 5). Differential abundance at the phylum level in the pre- and post-treatment states was similar in all experimental groups. However slight decrease of *Campylobacterota* and *Desulfobacterota* was detected at the end of the experiment in all three groups (Appendix A).

It was found that in the *Control* group during the experiment, there was a significantly increase in the *F/B* ratio by 2.27 times (*p* = 0.02). In the *CoQ10* group the *F/B* ratio remained unchanged (Table 3).

#### 2.5.3. The Alterations of the Gut Microbiota

Analysis of differential abundance performed with three independent algorithms implemented in microeco package showed that abundance of 63 genera differed significantly before and after treatment at least by one method in one experimental group (Appendix A). However, only 10 of these observations (3, 2 and 5 for *Control, Vehicle* and *CoQ10* groups, respectively) were supported by two or more algorithms and were considered for result interpretation. By this approach genera, differentially represented before and after treatment for *Control* group included *Turicibacter* (increased 10.4 times), *Erysipelatoclostridium* (increased 5.1 times), *Lachnospiraceae AC 2044 group* (decreased by 2.5 times); for *CoQ10* group included Ruminococcus (increased 2.4 times), *Dorea* sp. (increased from undetectable amounts to 0.36%), *Helicobacter* (decreased 2.8 times), *Lachnospiraceae AC 2044 group* (increased by 7.5 times), *Pygmaiobacter* (increased from undetectable amounts to 0.15%). However, except of *Helicobacter*, which mean abundance before treatment was 2.4% and decreased to 0.6%, all of these genera were minor and represented less than 1% of the community.

#### 2.5.4. Correlation Analysis between Taxonomic Composition and the Level of Metabolites of Gut Microbiota

Correlations analysis between gut microbiome and metabolic markers with following comparison of the groups showed a negative relationship with *Fecalibacterium* (r = −0.8131, *p* = 0.0004); in the *Vehicle* group with *Turicibacter* (r = 0.7799, *p* = 0.001), *Clostridium sensu stricto 1* (r = 0.7799, *p* = 0.001) in *control* group (Figure 6). A positive correlation was observed with the representation of *Erysipelatoclostridium* taxa (r = 0.561, *p* = 0.03), *Allobaculum* (r = 0.707, *p* = 0.0062) and negative with *Helicobacter* (r = −0.5774, *p* = 0.02) in the *CoQ10* group. And the level of methane in exhaled air in rats in the *CoQ10* group was negatively correlated with the abundance of the genus *Lachnospiraceae AC 2044 group* (r = −0.593, *p* = 0.02). Fecal acetate, propionate, and butyrate levels were negatively correlated with the abundance of *Turicibacter* (r = −0.7758, *p* = 0.002; r = −0.7104, *p* = 0.003; r = 0.84, *p* = 0.0001) and *Clostridium sensu stricto 1* (r = −0.731 *p* = 0.005; r = 0.6731, *p* = 0.006; r = −0.833, *p* = 0.0002). Fecal TMA levels were negatively correlated with *Allobaculum* (r = −0.5728, *p* = 0.035) in the *Control* group and *Lactobacillus* (r = −0.6137, *p* = 0.014) in the *CoQ10* group.

## 3. Discussion

For the first time, the data have been obtained on the ability of long-term administration of CoQ10 (21 days) to increase the formation of hydrogen and SCFA by the intestinal microbiota.

Coenzyme Q (CoQ) is a redox-active quinone derivative harboring a variable number of isoprene units that range from 7 to 12 in different species. In human tissues, CoQ10 is the predominant homolog, whereas in the rodent, CoQ9 is predominant, but homologous from Q8 to Q10 were detected in rat’s different tissues [13]. CoQ8 is the predominant homolog in bacteria [14,15], but in some bacteria CoQ10 is the sole respiratory quinone [16]. The possibility of the effect of injected CoQ10 on the level of CoQ8 in microbiota bacteria has not been determined in our experiments and we have not found an answer to this question in the literature. It was shown, that administration of CoQ10 in mice increase the concentration of it in the small intestine wall by 10-fold, but did not change the level of CoQ9 [17].

The reasons for greater hydrogen production during a standard fasting lactulose breath test may be: greater production of hydrogen by available species of microbiota, a relative increase in hydrogen-producing bacteria, reduced in situ hydrogen consumption in the intestinal lumen by hydrogenotrophic microorganisms, primarily methanogenic archaea and several others (e.g., *Helicobacter pylori*). No evidence is available that more H_2_ production occurs under the influence of CoQ10 against a background of unchanged microbiota composition. However, there is still no evidence for a possible effect of CoQ10 on increasing hydrogen-producing bacteria in the microbiota. Fecal 16S rRNA sequencing revealed an increase in the proportion of *Ruminococcus and Lachnospiraceae AC 2044 group* bacteria, which according to literature data are involved in H_2_ generation [18,19]. These bacterial species have [FeFe] hydrogenases required for H_2_ generation during substrate fermentation. At the same time, administration of CoQ10 caused suppression of the relative abundance of *Helicobacter* bacteria having [NiFe] hydrogenases. According to the HydDB database [20], bacteria possessing [NiFe]-type hydrogenases are able to utilize hydrogen. The increase in the ratio of hydrogen generating/ hydrogen utilizing bacteria can explain the increased production of hydrogen by gut microbiota against the background of CoQ10 intake.

The suppression of the activity of hydrogenotrophic, methanogenic microbiota microorganisms may be of significant importance in the increase in H_2_ formation under the influence of CoQ10. The rats used in our experiments were characterized by high content of methane in the air samples studied, and can be referred by this criterion to the group of highly methanogenic. It is known that H_2_ formed as a result of fermentation of mainly carbohydrates is used for acetate synthesis by acetogenic bacteria, for methane synthesis by methanogenic archaea (this appears to be the main consumer of H_2_), and for hydrogen sulfide synthesis by sulfate-reducing bacteria. The synthesis of one molecule of methane requires 4 molecules of hydrogen [19,21]. The increase in H_2_ production is demonstrated occurs simultaneously with a decrease in CH_4_ synthesis. Such explanation is supported by the recently published paper by Ruoud and colleagues [22]. The authors have showed that in cocultured of hydrogen producing *Christensenella minuta* and methane producing *Methanobrevibacter smithii* condition form flocks (very tight conenctions) in vitro and methanogenic *Methanobrevibacter* is able to utilise all the hydrogen produced by *Christensenella minuta*. It is unlikely that CoQ10, being a benzoquinone, directly affects the Archaea functions, since Archaea isoprenoid quinones contain Menaquinone (MK), which belongs to the naphthoquinones class [7].

The increase in the level of H_2_ in air samples is not only a biomarker of the enzymatic activity of the intestinal microbiota, but may also have an independent significance in increasing the antioxidant defense of the body. In the work of a group of Japanese authors led by Professor Shigeo Ohta, published in 2007, it was shown that molecular hydrogen is an antioxidant that can neutralize the most active forms of oxygen—hydroxyl radical (**•**OH) and peroxonitrite (ONOO^−^) [9]. In subsequent years, more than 2000 scientific articles were published, confirming the main finding of Professor Shigeo Ohta et al. and showing the positive effect of H_2_ introduced in the form of hydrogen saturated water as an antioxidant in ischemia/reperfusion of the brain, liver [23] and heart [24] and many other pathological conditions in the pathogenesis of which oxidative stress plays an important role [25,26]. The anti-inflammatory effects of molecular hydrogen have also been demonstrated [27,28].

Endogenous hydrogen produced by microbiota carbohydrate fermentation, is absorbed into the bloodstream and enters the liver at maximum concentration through the portal system. Nishimura and colleagues showed that the addition of a microbiota fermentation substrate (pectin) was accompanied by an increase in hydrogen production in rats. The authors found an increase in the concentration of H_2_ in the portal vein blood (4.8-fold with a range of absolute concentrations from 4.5 to 11.5 μmol/L) and total H_2_ output 15.5 times higher compared with controls [23].

There are reasons to compare the antioxidant effects of hydrogen and CoQ10, on models of liver ischemia/reperfusion, in the pathogenesis of damage of which oxidative stress plays a key role. In an experimental model of hepatic reperfusion ischemia in rats, it was shown that intraperitoneal injection of 10 mL/kg of hydrogen-enriched water reduced the production of proinflammatory cytokines and oxidative stress markers [29]. Administration of CoQ10 at a dose of 100 mg/kg/day for 10 days (3.3 times the dose compared to our study) also contributed to a reduction in the effects of hepatic ischemia/reperfusion [30].

In many bacteria H_2_ generation and synthesis of short-chain fatty acids occur in parallel [31]. The increase in the partial pressure of hydrogen in the gastrointestinal tract determines thermodynamically favorable conditions for SCFA production. And the formation of acetate and butyrate is more advantageous energetically than the formation of propionate at high hydrogen concentrations in the environment [32]. This was confirmed by our study. In the *CoQ10* group an increase in the rate of hydrogen formation, recorded after the lactulose load, took place an increase the concentration of acetate by 56% (*p* = 0.04) and butyrate by 126% (*p* = 0.04) in the feces with respect to the initial levels.

In the *CoQ10* group, animals showed a marked inhibitory effect on the growth of *Firmicutes* bacteria. Perhaps this is related to the reduction of TMA formation in the intestine, as *Firmicutes* bacteria are characterized by the presence of genes associated with the formation of TMA [33]. The reduction of TMA levels in the intestine (precursor of TMAO) may reflect an additional positive effect of coenzyme Q10 in relation to the risk cardiovascular diseases development.

In our study, oral administration of CoQ10 at a dose of 30 mg/kg/day for 21 days resulted in increased α- and β-diversity. Increased diversity is known to rise ecosystem stability and resilience [34] and serves as a marker of gut microbiota health and metabolic capacity [35]. Comparison of changes in the relative abundance of taxa at the phylum level revealed no changes in the experimental groups *Vehicle* and *CoQ10*, while in the *Control* group there was a marked increase in the ratio of *Firmicutes* to *Bacteroidota* phylum (*F/B*). *Firmicutes* and *Bacteroidota* are dominant phyla and a change in their ratio in the feces is considered by many researchers to be a marker of the metabolic health of the body. Increased *F/B* ratio is a sign of microbiota dysbiosis accompanying the development of such diseases as obesity, type 2 diabetes, arterial hypertension etc. Animals receiving CoQ10 showed a restrained increase in this index in our study. The positive effect of CoQ10 on the intestinal microbiota is consistent with the results of Bodkin [36], who showed that administration of CoQ10 reduced the degree of dysbiosis, prevented an increase in permeability of the intestinal wall and induced growth of “beneficial” lactobacilli in a model of hypoxia in newborn rats. CoQ10 in control rats produced the highest abundance of *Lactobacillus* species (95.3%), but this declined to 60.4% in rats with intermittent hypoxia. Low abundance phyla in CoQ10 in rats with intermittent hypoxia were *Firmicutes*, *Proteobacteria*, *Verrucomicrobia* and 2.8% unidentified organisms.

Several scientific papers confirm the effects of CoQ10 on the GI tract. An experimental model of ulcerative colitis showed antioxidant and anti-inflammatory effects of CoQ10 at a dose of 30 mg/kg by increasing the content of glutathione, catalase activity, reduced malonic dialdehyde level in the colon tissues [37]. Coenzyme Q was found to accumulate well in the intestinal walls throughout the intestine during radiation exposure in mice and protect the crypts from peroxidation and apoptosis [17]. Possibly, the increased activity of enzymatic antioxidants against the background of CoQ10 application may be partially related to the formation of endogenous hydrogen by the gut microbiota.

In the literature, there are indications of the ability of molecular hydrogen to enhance the effects of CoQ10 at the mitochondrial level. In the study of Gvozdjáková [12] the administration of hydrogen-saturated water *per os* on the second day led to the stimulation of mitochondrial respiratory electron chain function in rat cardiomyocytes, increased the level and rate of ATP production by Complex I and Complex II substrates, as well as to an increase in coenzyme Q formation in mitochondria. Combining results of our study with the results of Gvozdjáková we propose a hypothesis on the existence of positive feedback for the antioxidant effects of CoQ10.

An increase in the intestine concentration of orally administered CoQ10 is followed by the changes in composition of microbiota and increase in hydrogen production. The more of hydrogen are absorbed into the blood, reaches mitochondria of the different organ and tissues and facilitates production of ATP and antioxidant activity of CoQ10.

## 4. Materials and Methods

### 4.1. Animals and Experimental Design

All experimental procedures were carried out according to the Guide of the Care and Use of Laboratory Animals (8th edition) and other approved guidelines. The research protocol was approved by the Bioethics Commission of Moscow State University, Institute of Biology (Protocol 129-a, 12 May 2021).

The study was performed on 22 male Wistar rats (210–230 g), which were purchased from the nursery of laboratory animals branch “Stolbovaya”, Scientific Center for Biomedical Technologies of the Federal Medical and Biological Agency of Russia (Stolbovaya, Russia). The rats were kept under 12 h light/dark conditions with free access to water and food, humidity and temperature control was also performed. The adaptation period after transportation was at least 14 days.

The animals were randomized into three groups.

Group 1—*Control* (n = 6)—purified water intake; 

Group 2—*Vehicle* (n = 8)—solvent of the coenzyme Q10;

Group 3—*CoQ10* (n = 8)—coenzyme Q10 30 mg/kg/day.

All rats consumed a chow diet for laboratory animals without any restriction. The substances were applied intragastrically in rats daily for 21 days. Group 3 rats received coenzyme Q10 as part of “Kudesan forte”, *vehicle* (group 2) and purified water (group 1) were administered in an equivalent volume (not more than 1 mL). The same dose of coenzyme Q10 (30 mg/kg/day) was used in Ewees’ work [37]. The group *Vehicle* was introduced additionally due to the possible modulating effect of excipients on the intestinal microbiota.

Composition of “Kudesan Forte”, Rusfik LLC, Recordati S.p.A., Italy (1 mL): coenzyme Q10 60 mg, vitamin E 6.8 mg, purified water, macrogol glyceryl hydroxystearate, sodium benzoate, citric acid, ascorbyl palmitate.

The taxonomic composition of the gut microbiota, the level of H_2_, CH_4_ in the air sampling, the level of SCFAs in plasma and feces, and TMA in rat’s feces were evaluated at the beginning and end of the experiment. 

### 4.2. Determination of Gas Metabolites of Gut Microbiota (H_2_ and CH_4_) 

The metabolic activity of the gut microbiota was assessed by measuring the concentration of gaseous metabolites of the microbiota (H_2_ and CH_4_) in the total air sample (exhaled air + flatus). Measurements for each rat were performed in two stages:

#### 4.2.1. Determination of H_2_ and CH_4_ in the Rat Total Air Sample on the Regular Diet

Air sampling was performed at three time points (0 h, 3 h, 6 h) in rats without restricted access to water and food starting at 10:00 am. At the end of the measurements, the animals were seated in metabolic chambers for 12 h of fasting with free access to water. The next day, measurements were taken in the lactulose test.

#### 4.2.2. Determination of H_2_ and CH_4_ in the Rat Total Air Sample in the Lactulose Test

Air sampling was performed at 7 time points (0, 2, 4, 5, 6, 7, 8 h). The first measurement was performed on an empty stomach. Then, an aqueous solution of lactulose (FresenlusKabl IPSUM S.r.l., Vicchio, Italy) was administered intragastrically at a dose of 2.0 g/kg of rat weight. Then the rats were placed in individual cages until the next time point. Next, a lactulose-evoked concentration-time curve was plotted for H_2_ and CH_4_, and the area under the AUC_0–8 h_ curve (ppm*h) was calculated, which reflects the rate of gas generation during 8 h of measurements.

At the end of 21 day of the experiment, both steps of gas metabolite determination were repeated sequentially.

### 4.3. Experimental Setup for Sampling Total Air Samples of Rats

To accumulate hydrogen and methane released by the rat (exhaled air + flatus), an installation was designed in our modification, similar to the one presented in the work of Gumbman [38] (Figure 7).

The installation consisted of a glass desiccator with a volume of 1.8 L and a life support system. Two cartridges were sequentially built into the air recirculation system, one of which was filled with sodium lime to remove CO_2_, the second with hydrogel to absorb water condensate formed during the animal’s respiration. A U-shaped manometer filled with mineral oil was used to supply oxygen to the system. When O_2_ was consumed by the rat, the air pressure in the recirculation system decreased and then oxygen entered the system through a U-shaped pressure manometer. This ensured the oxygen concentration at the level of 19–22%, sufficient to ensure the vital functions of the animal. The time during which the rat was in the air recirculation system was calculated from the free volume of the desiccator and the rat’s air exchange rate from tabular values (27.27 ± 2.39 mL/min * 100 g) [39]. Air circulation and oxygen content control were provided by a device built at the Saint Petersburg State University of Aerospace Instrumentation (SUAI, St. Petersburg, Russia) and consisting of a pump and an oxygen sensor (OKSIK-3, Russia) for monitoring the oxygen level in the system. This device provides an air flow at the 400 mL/min rate and estimates the oxygen concentration in the flow in the range of 0–100% with a resolution of 0.1%. Supply and exhaust ventilation were provided through 2 stainless steel tubes inserted into the rubber stopper of the desiccator.

After holding the rat in the desiccator for an estimated time, an air sample in the volume of 1 mL was taken with a Hamilton gas-tight glass syringe through a valve built into the recirculation tube system.

### 4.4. Measurement of H_2_ and CH_4_ Concentrations in the Total Air Sample

The concentration of H_2_ and CH_4_ in the air sample was measured by gas chromatography (TRILyzer mBA-3000, Taiyo Instruments Inc., Japan). As a carrier gas, we used artificial air gas of “A” quality (TU 6-21-5-82): oxygen (O_2_) (20.9%) + nitrogen (N_2_) (the rest). The measurements were carried out at a carrier gas flow rate in the chromatograph system of 30 mL/min and a column temperature of 50 °C. Gas air mixtures with 5 ppm (low point) and 50 ppm (high point) volume concentrations of H_2_ and CH_4_ were used to calibrate the gas chromatograph. Before starting measurements of animal samples, the background values of H_2_ and CH_4_ concentrations in the ambient air were measured. Background values were subtracted from the values obtained from the analysis of the total air sample from the animal.

### 4.5. Collection of Fecal and Plasma Samples

Fecal and blood samples were taken twice: at the beginning and at the end of the experiment. Fecal samples were collected in individual Eppendorf-type tubes, quickly frozen in liquid nitrogen and stored at −70 °C until analysis. For sequencing, fecal samples were taken and stored at −70 °C until analysis without prior freezing in liquid nitrogen.

Blood sampling was performed from the sublingual vein after preanesthesia of the animal with isoflurane using Pureline M6000 inhalation anesthesia machine (Supera Anesthesia Innovations, Estacada, OR, USA). Blood samples of 1 mL were collected in Eppendorf-type tubes with heparin. Samples were then centrifuged at 5000× *g* for 3 min. 0.3–0.5 mL of plasma was collected, frozen, and stored at −70 °C until further analysis.

### 4.6. Quantification of SCFA in Rat Feces and Blood

Quantification of rat feces and blood SCFA was performed by nuclear magnetic resonance (NMR) spectroscopy. 

#### 4.6.1. Fecal Sample Preparation

Sample of fecal matter (0.1–0.5 g) was mixed with 1× PBS buffer (Amresco, OH, USA) in a ratio of 1:3 and homogenized using a PrecellysEvolution homogenizer (Bertin Technologies, Montigny-le-Bretonneux, France) in tubes containing zirconium oxide beads (3 times for 10 s). After that it was plased on an orbital shaker for 30 min and centrifuged at 16,000× *g* for 15 min at 4 °C.

The supernatant was filtered through Amiconultracellulose cartridges with a cut-off molecular weight of 10 kDa (Merck Millipore, NJ, USA). The filtrate was dried in a vacuum concentrator (Eppendorf Concentrator plus™) for 3 h until the solvent completely evaporated, and then it was freeze-dried using FreeZone 2.5 freeze dry system (Labconco, Kansas City, MO, USA) for 3 h and kept at −60 °C until measurements. 

#### 4.6.2. Blood Plasma Sample Preparation

Blood plasma samples were prepared according to the recommendations [40]. 200 µL of blood plasma was mixed with methanol (200 µL) and chloroform (200 µL), precooled to −20 °C. The samples were thoroughly vortexed, kept for 10 min at 5 °C, and then centrifuged at 16,000× *g* for 30 min at 5 °C. The upper water-methanol layer was carrefully separated and dried on a vacuum concentrator (Eppendorf Concentrator plus™) for 6 h until the solvent completely evaporated, and then freeze-dried using FreeZone 2.5 freeze dry system (Labconco, Kansas City, MO, USA) for 3 h and stored at −60 °C until measurements. 

#### 4.6.3. NMR Spectra Mesurements

The dried extracts were dissolved in 550 μL of deuterated phosphate buffer (20 mM, pH 8.0) containing sodium azide (2 mM) as a preservative against bacteria and DSS-d6 (0.1 mM) as an internal standart. After centrifugation (10 min, room temperature) samples were placed into standard 5 mm NMR tubes. The spectra were acquired on AVANCE Neo 700 MHz NMR spectrometer (Bruker BioSpin, Reinstetten, Germany) equipped with a Prodigy triple resonance cryoprobe (Bruker, Reinstetten, Germany). The spectra were acquired at 25 °C using the noesypr1d pulse sequence with the following parameters: 131,072 data points, 4 dummy scans, 400 scans for fecal and 600 for plasma samples, 19.8364 ppm spectral width; 4.7 s accumulation time, 6.0 s relaxation delay between scans.

#### 4.6.4. Identification and Quantification of Metabolites

Spectra processing and metabolite identification were performed in the Chenomx v9 (Chenomx Inc., Edmonton, AB, Canada) program based on the built-in database, as well as the Human Metabolome Database [41]. The assigments was confirmed by adding reference SCFA compounds. The concentrations of metabolites were measured related to DSS peak area in Chenomx v9 (Chenomx Inc., Edmonton, AB, Canada). 

### 4.7. HPLC Analysis

The fecal samples were homogenized in 96% C_2_H_5_OH (ratio 1:4) by ultrasound homogenizer. 100 μL homogenate was transferred in an individual eppendorf-type tube for further extraction. N-Hexan (250 μL) was added to the fecal samples, ethanol (200 μL) and n-hexane (500 μL) were added to 100 μL of plasma and were shaken for 10 min, then centrifuged at 5000× *g* for 5 min. The upper hexan layer was collected and the extraction was repeated. The total extract amount was evaporated and dissolved in 100 μL of 96% ethanol. CoQ10 levels were measured using high-performance liquid chromatography with electrochemical detection (Environmental Sciences Associate, Inc., Chelmsford, MA, USA): model 580 pump and electrochemical detector “Coulochem II”, in isocratic mode on Luna column 150 × 4.6 mm with sorbent C18 (5 μm) at a flow rate of the eluent of 1.3 mL/min. The mobile phase was 0.3% of NaCl in the mixture ethanol-methanol-7% HClO_4_ (970:20:10). Electrochemical detection carried out the oxidizing mode using an analytical cell (model 5011) with the voltage of −50 mV and +350 mV applied for the two pairs of electrodes, correspondingly. Chromatographic data registration and processing carried out by using of the Environmental Sciences Associate Inc software (Chelmsford, MA, USA). The extract was analyzed both before and after the complete recovery of ubiquinol (by adding the solution of sodium tetrahydroborate in ethanol).

### 4.8. Analysis of Microbial Community Composition

#### 4.8.1. DNA Extraction and High-Throughput 16S Sequencing

Extraction of DNA from feces was performed with QIAampPowerFecal Pro DNA Kit (Qiagen, Hilden, Germany), according to manufacturer’s instructions. Quantity of DNA was assessed using Qubit™ 4 fluorometer (Thermo Fisher Scientific, MA, USA). 1–2 ng of DNA was used for library preparation.

Libraries of 16S rRNA V4 hypervariable region were prepared using a two-step PCR strategy as described earlier [42]. Two library replicates were prepared for each sample. The first round of PCR was performed using qPCRmix-HS SYBR (Evrogen, Moscow, Russia) with the following primers at a concentration of 0.25 μM: V4_515F TCGTCGGCAGCAGCGTCAGATGTGTGTGATAAGAGACAG [NN] GTGBCAGCMGCCGCGGTAA and V4_805R GTCTCTCGTGGTGGCTCGGAGATGTGTGTAGATAAGACAG [NN] GACTACNVGGGTMTCTAATCC, where the first part corresponded to a partial Illumina TruSeq adapter, [NN] corresponds to 1–3 nt spacer of degenerate heterogeneity, and the last part corresponds to primers 515F [43] and Pro-mod-805R [44], respectively. After first step diluted amplification mixture was used as a matrix for the second PCR. The second PCR was performed using ScreenMix-HS (Evrogen, Moscow, Russia) using the following primers at a concentration of 0.5 µM: R1TM AATGATACACGGCGACCACCGAGATCTACACAXXXXXXCGTCGGCAGCGTC and R2TM CAAGCAGAAGACGGCATACGAGATXXXXXXGTCTCTCGTGGTGGCTCGG, where the first part corresponded to P5 or P7 Illumina oligonucleotides, XXXXXX corresponded to 6-nt index sequences, and the last part corresponded to Illumina TruSeq partial adaptors annealing to the tail of the first PCR primers. Amplification was performed on a Veriti thermal cycler (Applied Biosystems, MA, USA). The resulting libraries were checked on an agarose gel and pooled equimolarly. The final pool was purified using the QIAquick Gel Extraction Kit (Qiagen, Germany) according to the manufacturer’s protocols.

#### 4.8.2. Primary Analysis of Sequencing Data

Sequencing was performed using the MiSeq™ Personal Sequencing System (Illumina, San Diego, CA, USA) with 2 × 156 bp paired end. The ‘Trim reads’ tool in CLC Genomics Workbench 20.0 (Qiagen, Hilden, Germany) was used to efficiently filter out and remove nucleotides corresponding to primers 515F and Pro-mod-805R. Demultiplexing was performed using the deML package [45], using zero available mismatches option. Then, read pairs were processed with DADA2 pipeline [46], according to published.

#### 4.8.3. Bioinformatics

High quality read pairs were processed with DADA2 pipeline [44], according to published protocol [46]. Taxonomy of amplified sequence variants (ASVs) was determined with naive Bayesian classifier using the Silva138 database [47]. For further analysis reads were rarefied using the phyloseq package [48]. The obtained ASV reference sequences, sample metadata, abundance tables, and taxonomy were imported into the phyloseq package, and all further operations were performed with the phyloseq object. Community differences in each group were assessed using the adonis and betadisper functions of vegan package [49]. Analysis of differentially abundant genera was performed with ANCOM-BC [50], DESeq2 [51] algorithms and the Wilcoxon rank sum test, implemented in microeco package (multivariate analysis of variance using distance matrices) [52]. Analysis of variance was performed using the betadisper function.

### 4.9. Statistical Analysis

Statistical analysis of the results was performed using GraphPadPrism 8 software (GraphPad, San Diego, CA, USA). The normality of the distribution was checked using the Shapiro-Wilk test to compare the mean values of a single index in more than two samples; for pairwise comparison of groups, the paired and unpaired t-test for dependent and independent samples was used, respectively. For pairwise comparisons of groups with non-normal distributions, the Wilcoxon test was used for dependent samples and the Mann-Whitney test for independent samples. Correlations were calculated using Pearson rank correlation coefficient. Exclusion of statistical outliers was performed using ROUT criterion with Q not exceeding 1%. Differences were considered statistically significant at *p* < 0.05. All data are presented as mean ± standard error of the mean (Mean ± SEM).

## Figures and Tables

**Figure 1 pharmaceuticals-16-00686-f001:**
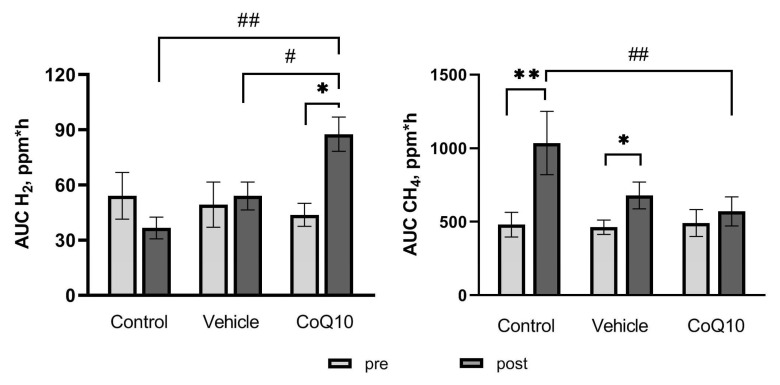
Area under the curve (AUC) of concentration of H_2_ and CH_4_ in rats (ppm*h) in the lactulose test. * *p* < 0.05 and ** *p* < 0.01—within-group comparisons, # *p* < 0.05 and ## *p* < 0.01—between-group comparisons.

**Figure 2 pharmaceuticals-16-00686-f002:**
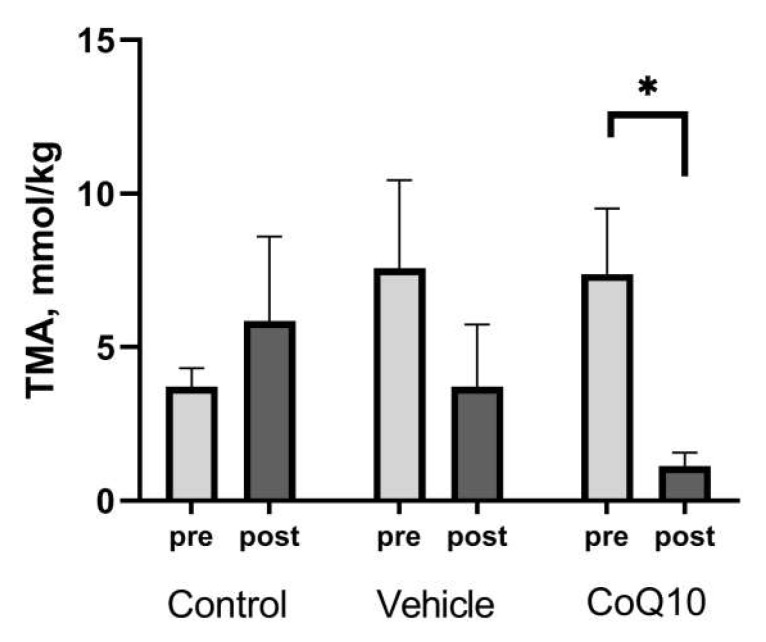
Trimethylamine (TMA) concentration in the feces of rats. * *p* < 0.05 significant differences in the «pre-post» condition.

**Figure 3 pharmaceuticals-16-00686-f003:**
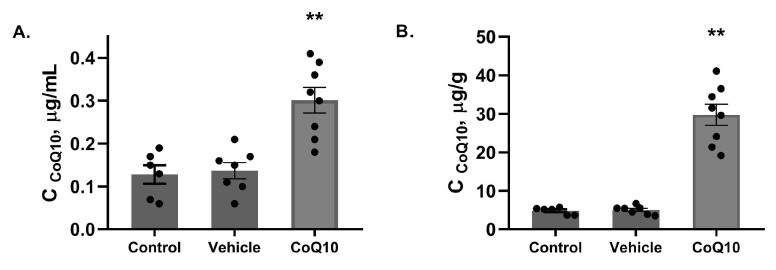
The concentration of CoQ10 in rats at the end of the experiment in plasma (**A**), in feces (**B**). ** *p* < 0.01, *CoQ10 vs. Control, Vehicle*—significant differences in the «post» condition.

**Figure 4 pharmaceuticals-16-00686-f004:**
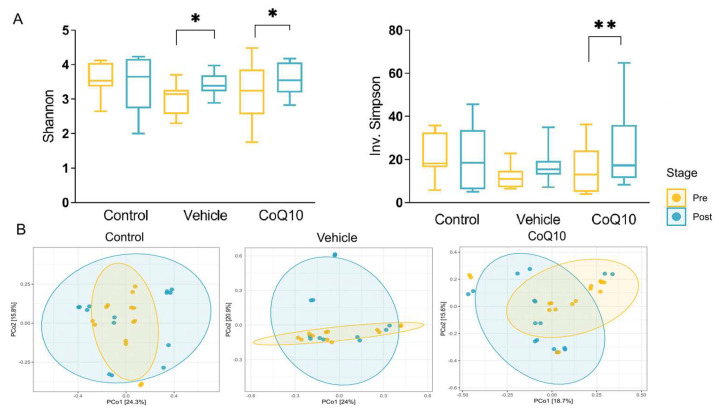
Visualization of alpha−diversity analysis of the microbiota in the «pre−post» experimental groups (**A**), Principal coordinate analysis (PcoA) of bacterial communities in the «pre−post» experimental groups (**B**). ** *p* < 0.01; * *p* < 0.05 statistically significant differences in the «pre−post» condition.

**Figure 5 pharmaceuticals-16-00686-f005:**
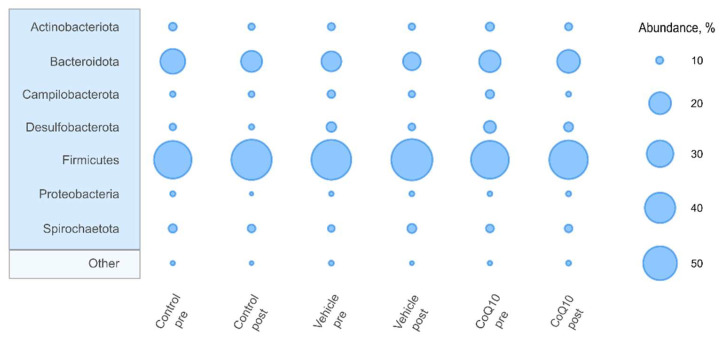
Relative abundance of top 7 phyla at pre- and post-treatment stages in all experimental groups.

**Figure 6 pharmaceuticals-16-00686-f006:**
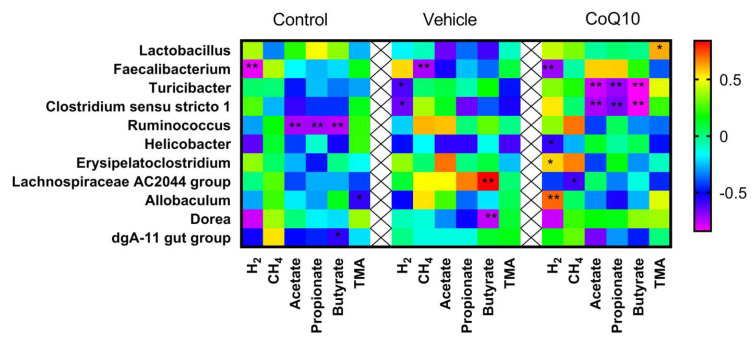
Correlation analysis between taxonomic composition and levels of gut microbiota metabolites in the experimental groups. ** *p* < 0.01; * *p* < 0.05.

**Figure 7 pharmaceuticals-16-00686-f007:**
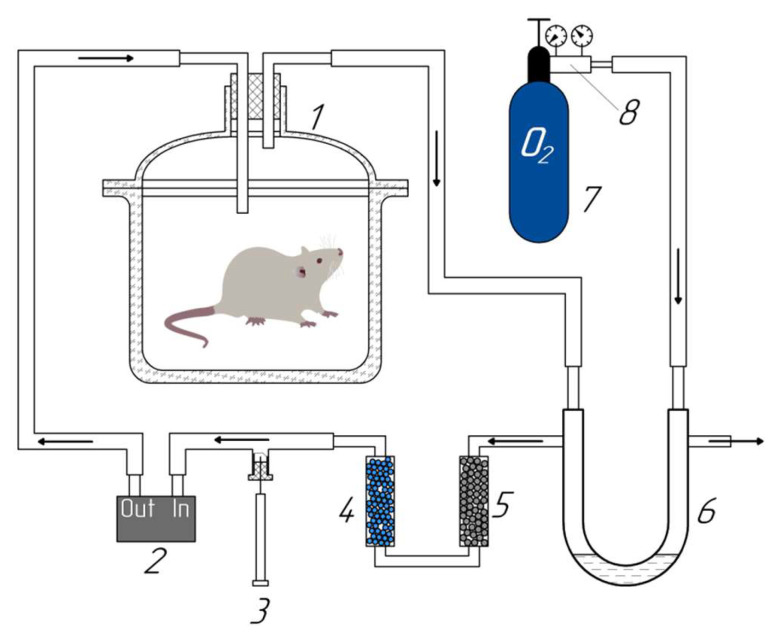
Diagram of the experimental setup for collecting the total air sample in the experiment. 1—desiccator, 2—air pump, 3—glass syringe, 4—moisture absorbent (hydrogel), 5—carbon dioxide absorbent (soda lime), 6—manometer, 7—oxygen cylinder, 8—gas pressure reducer.

**Table 1 pharmaceuticals-16-00686-t001:** The level of gas biomarkers of the gut microbiota (H_2_ and CH_4_) in the air sample at the beginning and at the end of the experiment in rats.

	*Control*			*Vehicle*			*CoQ10*		
	Pre	Post	*p*-Value	Pre	Post	*p*-Value	Pre	Post	*p*-Value
H_2_, ppm	12.8 ± 2.2	11.8 ± 2.3	0.6	9.1 ± 3.2	9.2 ± 1.3	0.4	12.3 ± 2.6	13.6 ± 3.1	0.5
CH_4_, ppm	88.7 ± 9.5	96.5 ± 14.6	0.6	88.3 ± 14.0	74.5 ± 12.2	0.4	99.0 ± 15.7	92.7 ± 15.5	0.8
CH_4_/H_2_	8.0 ± 1.8	12.4 ± 4.1	0.3	11.9 ± 1.9	9.2 ± 1.9	0.2	10.1 ± 2.4	5.9 ± 1.5	0.1

**Table 2 pharmaceuticals-16-00686-t002:** The concentration of short-chain volatile fatty acids (SCFAs) in rats in the experiment. A—in feces, B—in plasma.

		*Control*			*Vehicle*			*CoQ10*		
		Pre	Post	*p*-Value	Pre	Post	*p*-Value	Pre	Post	*p*-Value
A.	Acetic acid, mmol/kg	5.5 ± 1.5	5.7 ±1.2	0.8	3.9 ± 1.6	4.5 ± 1.7	0.8	4.9 ± 1.1	7.5 ± 1.8	* 0.04
	Propionic acid, mmol/kg	2.50 ± 0.81	2.62 ± 0.60	0.8	1.49 ± 0.66	1.73 ± 0.64	>0.9999	2.29 ± 0.50	3.42 ± 0.75	0.09
	Butyric acid, mmol/kg	0.44 ± 0.12	0.49 ± 0.11	0.4	0.31 ± 0.15	0.36 ± 0.13	0.7	0.28 ± 0.06	0.63 ± 0.18	* 0.04
	Total SCFA, mmol/kg	8.4 ± 2.4	8.8 ± 1.9	0.7	5.7 ± 2.4	6.6 ± 2.3	0.8	6.8 ± 1.6	11.1 ± 2.5	* 0.02
B.	Acetic acid, μmol/L	199 ± 21	215 ± 6	0.6	240 ± 19	235 ± 11	0.7	189 ± 22	246 ± 33	0.08
	Propionic acid, μmol/L	27.2 ± 6.6	23.0 ± 6.7	0.4	34.3 ± 8.1	32.1 ± 7.8	0.4	30.7 ± 9.8	36,6 ± 9.8	0.4
	Butyric acid, μmol/L	8.9 ± 1.1	7.8 ± 1.9	0.6	8.4 ± 0.7	8.8 ± 1.0	0.7	7.0 ± 0.8	7.0 ± 0.6	0.98
	Total SCFA, μmol/L	233 ± 27	244 ± 8	0.7	282 ± 25	276 ± 18	0.7	227 ± 31	290 ± 43	0.1

* *p* < 0.05 significant differences in the «pre-post» condition.

**Table 3 pharmaceuticals-16-00686-t003:** *Firmicutes/Bacteroidota* ratio in the experimental groups.

	Pre	Post	*p*-Value
*Control*	2.76 ± 0.41	6.26 ± 1.46	** *p* = 0.0017
*Vehicle*	5.20 ± 1.25	8.16 ± 1.71	*p* = 0.09
*CoQ10*	4.60 ± 0.76	3.93 ± 0.64	*p* = 0.25

** *p* < 0.01 significant differences in the «pre-post» condition.

## Data Availability

The data presented in this study is contained within the article. Individual values are available from the corresponding authors upon reasonable request.

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
