# Peer review of "Effects of Coenzyme Q10 on the Biomarkers (Hydrogen, Methane, SCFA and TMA) and Composition of the Gut Microbiome in Rats"

_pharmaceuticals, 2023, doi:10.3390/ph16050686_

Round 1
Reviewer 1 Report
Dear authors,
I am very satisfied with the outcome of this Ms, and I have only a few comments.
Abstract
Can you please write the full name of coenzyme Q10 for the first time then its abbreviation.
Introduction:
Can you please delete the first and the second sentences, as they look like instructions.
Materials and Method:
Kindly, can you please write the reference of the dose of CoQ10.
Page 13, line 488, correct methabolites to metabolites.
Reviewer 2 Report
The manuscript is a research article on the Effects of Coenzyme Q10 on the biomarkers and composition of the gut microbiome in rats. It is considered that the manuscript will be interest to readers as the results provides basic scientific information about that orally administered CoQ10 is followed by the increase in hydrogen production of modulated microbiota and facilitates production of ATP and antioxidant activity of CoQ10.
The manuscript is written in logical and well-ordered sentences. However, several points should be modified to be accepted in the Journal.
1. In the Materials and Methods section, an explanation is needed to understand the difference between the N number of the control group and the other groups.
2. Animal experiment must be approved by the Animal Ethics Committee. The contents of the institution's ethics committee approval must be stated in the manuscript.
3. In Result 2.6; The composition of each group's phylum will be different. Although outlined, it is need comparative data representing the overall composition of the phylum for each group.
4. In 2.6.3., there is no data on the overall change, and only a few are selected and described. A specific description of the result is required.
Reviewer 3 Report
In this manuscript, the authors administered CoQ10 orally to mice to achieve pharmacological of effects contributes to the creation of elevated concentrations of CoQ10 in the intestinal lume, which could have an effect on the gut microbiota and the levels of biomarkers it produces. These experiments demonstrated the levels of hydrogen, methane and SCFA was markedly increased, while the concentrations of TMA was significantly decreased. In my opinion, this manuscript can be considered for publication in Pharmaceutics after revisions.
1. The authors found that after feeding CoQ10, the gut microbiota changed, resulting in hydrogen production. The evidence was needed to provide on whether CoQ10 induces hydrogen production by related bacteria in vitro.
2. Related studies on the use of hydrogen in anti-inflammatory therapy are suggested to refer (Adv. Healthcare Mater. 2019, 8: 1900463; Nat. Commun. 2022, 13: 5684).
Round 2
Reviewer 2 Report
Appropriately corrected for all queries. I agree to have this manuscript published in the journal of Pharmaceuticals. Thank you.